# Genetic and Molecular Characterization of the Immortalized Murine Hepatic Stellate Cell Line GRX

**DOI:** 10.3390/cells11091504

**Published:** 2022-04-30

**Authors:** Sarah K. Schröder, Herdit M. Schüler, Kamilla V. Petersen, Cinzia Tesauro, Birgitta R. Knudsen, Finn S. Pedersen, Frederike Krus, Eva M. Buhl, Elke Roeb, Martin Roderfeld, Radovan Borojevic, Jamie L. Almeida, Ralf Weiskirchen

**Affiliations:** 1Institute of Molecular Pathobiochemistry, Experimental Gene Therapy and Clinical Chemistry (IFMPEGKC), RWTH University Hospital Aachen, D-52074 Aachen, Germany; saschroeder@ukaachen.de (S.K.S.); frederike.krus@rwth-aachen.de (F.K.); 2Institute of Human Genetics, University Hospital of RWTH Aachen University, D-52074 Aachen, Germany; hschueler@ukaachen.de; 3Department of Molecular Biology and Genetics, Aarhus University, DK-8000 Aarhus, Denmark; kvp@clin.au.dk (K.V.P.); brk@mbg.au.dk (B.R.K.); fsp@mbg.au.dk (F.S.P.); 4Department of Clinical Medicine, Aarhus University, DK-8000 Aarhus, Denmark; 5VPCIR Biosciences ApS., DK-8000 Aarhus, Denmark; ct@vpcir.com; 6Electron Microscopy Facility, Institute of Pathology, RWTH Aachen University Hospital, D-52074 Aachen, Germany; ebuhl@ukaachen.de; 7Department of Gastroenterology, Justus Liebig University, D-35392 Giessen, Germany; elke.roeb@innere.med.uni-giessen.de (E.R.); martin.roderfeld@innere.med.uni-giessen.de (M.R.); 8Institute of Biomedical Sciences, Federal University of Rio de Janeiro, Rio de Janeiro 21941-590, Brazil; rrborojevic@gmail.com; 9National Institute of Standards and Technology, Biosystems and Biomaterials Division, Gaithersburg, MD 20899, USA; jamie.almeida@nist.gov

**Keywords:** hepatic stellate cells, myofibroblasts, retinol, cell line, authentication, cross-contamination, STR profiling, genotyping, 3R principle, liver

## Abstract

The murine cell line GRX has been introduced as an experimental tool to study aspects of hepatic stellate cell biology. It was established from livers of C3H/HeN mice that were infected with cercariae of *Schistosoma mansoni*. Although these cells display a myofibroblast phenotype, they can accumulate intracellular lipids and acquire a fat-storing lipocyte phenotype when treated with retinol, insulin, and indomethacin. We have performed genetic characterization of GRX and established a multi-loci short tandem repeat (STR) signature for this cell line that includes 18 mouse STR markers. Karyotyping further revealed that this cell line has a complex genotype with various chromosomal aberrations. Transmission electron microscopy revealed that GRX cells produce large quantities of viral particles belonging to the gammaretroviral genus of the *Retroviridae* family as assessed by next generation mRNA sequencing and Western blot analysis. Rolling-circle-enhanced-enzyme-activity detection (REEAD) revealed the absence of retroviral integrase activity in cell culture supernatants, most likely as a result of tetherin-mediated trapping of viral particles at the cell surface. Furthermore, staining against schistosome gut-associated circulating anodic antigens and cercarial O- and GSL-glycans showed that the cell line lacks *S. mansoni*-specific glycostructures. Our findings will now help to fulfill the recommendations for cellular authentications required by many granting agencies and scientific journals when working with GRX cells. Moreover, the definition of a characteristic STR profile will increase the value of GRX cells in research and provides an important benchmark to identify intra-laboratory cell line heterogeneity, discriminate between different mouse cell lines, and to avoid misinterpretation of experimental findings by usage of misidentified or cross-contaminated cells.

## 1. Introduction

In the healthy liver, quiescent hepatic stellate cells (HSC) represent 5–8% of the total number of liver cells [1]. They contain numerous fat vacuoles that store vitamin A in the form of retinyl esters. Upon liver injury, resting HSC become activated and acquire a myofibroblast (MFB) phenotype in a process called transdifferentiation [2]. The transited phenotype is highly proliferative and expresses large quantities of extracellular matrix components including collagen. This temporal sequence of molecular events associated with myofibroblastic differentiation can be replicated in vitro by culturing freshly isolated primary HSC on uncoated tissue culture plastic. During prolonged culturing, the cells progressively lose their vitamin A, become positive for α-smooth muscle actin (α-SMA), and acquire a spread fibroblastic morphology [3]. Therefore, cultured primary HSC are common in vitro models used in investigations addressing issues of HSC biology and function. However, the limited supply and ever-increasing demand for these cells, combined with the additional tightening of formal standards in animal welfare policy, make working with these primary cells increasingly difficult. Therefore, many investigators have implemented continuous growing HSC lines from mouse, rat, and humans [4].

The continuous HSC line GRX was originally established from murine livers harboring fibrotic granulomas experimentally induced by transcutaneous penetration of 40 cercariae of *Schistosoma mansoni* in C3H/HeN mice [1]. These adherent cells are highly proliferative, have a fibroblastic morphology with nucleolated subspherical nuclei, which become elongated in cells that grew in multilayers. In the initial report, GRX cells were shown to produce retrovirus particles and positive for collagen types I, III, and IV, laminin and fibronectin [5]. In addition, a rough karyotype analysis in an uncloned population of GRX cells showed that these cells were aneuploid, with only slight deviation of normal chromosome values expected in mouse cells. As such, GRX cells display typical myofibroblastic characteristics and are considered to originate from liver connective tissue cells, namely HSCs. Previous in vitro work has shown that GRX cells can acquire a fat-storing phenotype corresponding to HSCs when treated with retinol, insulin, and indomethacin [6,7]. Based on their capacity to accumulate intracellular lipids, it was proposed that these cells are not directly involved in energy storage but represent a specialized cell population involved in storage and control of homeostasis of lipid-soluble substances [8]. The finding that retinol further suppressed proliferation, reduced collagen expression, increased adhesion and consequent spreading on the substrate suggests that retinol can induce re-programming by which these cells acquire a vitamin-A-induced non-proliferative, resting, fat-storing (lipocyte) phenotype, comparable to those characteristic of HSCs [9]. In line with this is the finding that the myofibroblastic phenotype correlated with increased collagen synthesis, whereas in the lipocyte phenotype collagen synthesis is markedly decreased [10]. Supplementary investigations further demonstrated that these immortalized cells behave very similar to primary cultures of HSCs, with respect to retinol uptake and release [11]. Like other HSC lines established from mice, rats, and humans, GRX cells have become a useful experimental tool used in different types of in vitro studies investigating aspects of retinoid metabolism, extracellular matrix biology, gene regulation, cytokine production and signaling, drug effects, and many other issues related to HSC biology [4]. Since the first report of this cell line in 1985 [1], GRX cells have been experimentally used or discussed as a suitable in vitro model for studying cellular and molecular features of HSC in approximately 70 studies from around world (Appendix A, Figure 1).

In addition, many abstracts report preliminary results in these cells, suggesting that the usage of these cells will increase dramatically during the next years.

However, the usage of cell lines is, in general, prone to misidentification, and with repeated passaging might provoke genotypic, karyotypic, or phenotypic drifts, which may result in cellular sublines that are phenotypically or genetically heterogenous [4]. Consequently, experimental findings obtained with GRX cells usually require validation in primary cultured HSCs.

Nevertheless, scientists are encouraged to implement the 3R principle (replacement, reduction, and refinement), an ethical framework proposed by Russell and Burch in 1959 [12]. In this regard, the usage of established cell lines is one possibility to limit animal experimentation [13].

With this in mind, we characterized the genetic profile of GRX cells by using short tandem repeat (STR) genotyping, standard karyotyping, and expression analysis of some markers specific for HSCs. By use of these cytogenetic and molecular methods, we could demonstrate that GRX comprise a complex karyotype with various structural abnormalities and chromosomal arrangements. However, we were able to define an invariant STR signature that allows proper cell authentication. Moreover, we show that GRX cells produce retroviral particles but lack *S. mansoni*-specific glycostructures.

## 2. Materials and Methods

### 2.1. Literature Search

Research papers using of GRX cells were identified by searching PubMed [14] or the Google search engine [15] with search terms “GRX cells” or “GRX and hepatic”. Papers that used GRX as an abbreviation for “glutathione/glutaredoxin”, “GSH reductase (GRx)”, “Glutaredoxin”, “GRX robotic”, “GRX-MCiPS4F-A2”, “Precision GRX”, “CorPath GRX”, “Green Prescription (GRx)”, ”GRX proteins”, “arsC (grx)”, “grepafloxacin (GRX)”, and “granulitoxin” were excluded. Moreover, articles found because of including “grx” in email addresses were eliminated. The final list of peer-reviewed papers that used the GRX cell line is given in Appendix A.

### 2.2. Cell Culture

GRX cells were obtained from the Banco de Células do Rio de Janeiro, Rio de Janeiro, Brazil (BCRJ code: 0094) and routinely cultured in Dulbecco’s modified Eagle’s medium (DMEM) supplemented with 10% fetal bovine serum, 2 mM L-glutamine, 1 mM sodium pyruvate, and 1× penicillin/streptomycin. Cells were subcultured at a ratio of 1:8 every third day. All experiments depicted were done with cells between passage 4 and passage 8 after receiving the cells from the cell bank repository. The human HSC cell line LX-2 [16] and rat HSC line HSC-T6 [17] were maintained in the same medium, while the medium for rat HSC line CFSC-2G [18,19] was further supplemented with 1% non-essential amino acids (#11140-35, Gibco, ThermoFisher Scientific, Schwerte, Germany). The hepatocytic AML12 cell line established from livers of transgenic mice overexpressing transforming growth factor-α hepatocytes of CD1 mouse strain [20] was obtained from the American Type Culture collection (# CRL-2254, ATCC, Manassas, VA, USA). The murine breast cancer cell lines 4T1 and EO771 were cultured as previously described [21]. All cells were routinely tested for mycoplasma infection by using the Venor^®^GeM OneStep Mycoplama detection kit for conventional PCR kit system following essentially the instructions of the manufacturer (Minerva Biolabs, Berlin, Germany).

### 2.3. Karyotyping

GRX cells were detached from the petri dish by Accutase cell detachment solution (Merck, Sigma-Aldrich, Taufkirchen, Germany) and transferred in T25 flasks and cultured for two days. The cultures were harvested following standard protocols including treatment with colcemid (10 µg/mL), hypotonic KCl solution (0.4%), and fixative (methanol: acetic acid (3:1)). G-banding by pancreatin staining with Giemsa (GPG) was performed following standard procedures. Cytogenetic analysis was done using an Axioplan 2 imaging microscope (Carl Zeiss, Oberkochen, Germany). For conventional chromosome analysis, five cells were analyzed by light microscopy. Karyotyping was done by using the IKAROS karyotyping system, version 5.9.1 (MetaSystems, Altlussheim, Germany). With this method, an average banding resolution of 250 bands was achieved. Aberrations were described according to the guidelines proposed by the International Committee on Standardized Genetic Nomenclature for Mice that were revised in September 2021 [22].

### 2.4. Short Tandem Repeat (STR) Profiling

STR profiling of GRX cells was done by using the 18 mouse consensus markers proposed by the Consortium for Mouse Cell Line Authentication for murine STR profiling, [23,24]. Fragment analysis was done on an ABI3130 Genetic Analyzer (Life Technologies, Darmstadt, Germany) and resulting data were analyzed with GeneMapper software v5 (Applied Biosystems, ThermoFisher Scientific). STR profiles were compared to published profiles in the Cellosaurus database by using the STR similarity search tool (CLASTR 1.4.4) at the Expasy server by using the default settings [25].

### 2.5. Electron Microscopic Analysis

GRX cells were scraped from the cell plate and individual cells separated by intensive pipetting. Thereafter, the cell suspension was fixed in 1× phosphate buffered saline containing 3% glutaraldehyde. After washing in 0.1 M Soerensen’s phosphate buffer (Merck, Darmstadt, Germany), the samples were post-fixed in 1% osmium tetroxide (OsO_4_) (Roth, Karlsruhe, Germany) solved in 25 mM sucrose buffer (Merck) and dehydrated by ascending ethanol series (30%, 50%, 70%, 90%, and 100%) for 10 min each. The last step was repeated three times. Subsequently, dehydrated specimens were incubated in propylene oxide (Serva, Heidelberg, Germany) for 30 min, in a mixture of Epon resin (Serva) and propylene oxide (1:1) for 1 h, and finally, in pure Epon for 1 h. Epon polymerization was performed at 90 °C for 2 h. Finally, ultrathin sections (70–100 nm) were cut with an ultramicrotome (Reichert Ultracut S, Leica, Wetzlar, Germany) by using a diamond knife (Diatome Ltd., Nidau, Switzerland) and picked up on Cu/Rh grids (HR23 Maxtaform, Plano GmbH, Wetzlar, Germany). Contrast was enhanced by staining with 0.5% uranyl acetate and 1% lead citrate (both Science Services, Munich, Germany). Samples were viewed without additional contrast staining at an acceleration voltage of 60 kV by using a Zeiss Leo 906 (Carl Zeiss) transmission electron microscope. Pictures were acquired at magnifications of 6000× to 100,000×.

### 2.6. Analysis of Potential Schistosoma Mansoni Load

Cells were stained by immunocytochemistry for *S. mansoni* specific glycostructures by using antibodies 54-4C2 [26] and 128-1E7 [27]. Positive controls used in this analysis were prepared from the gut of the adult worm or eggs in hepatic granuloma of bisex-infected 8-week-old mice from previous experiments. The respective animal experiments from this previous study have been done in accordance with the European Convention for the Protection of Vertebrate Animals used for experimental and other scientific purposes and have been approved by the Regional Council Giessen (approval number GI 20/10 Nr. G 44/2019). In addition, we analyzed mRNA expression of the *S. mansoni* specific glycoproteins IPSE/alpha 1, Kappa-5 and Omega-1 by using primer pairs: IPSE-alpha 1-for: 5′-GCG TTG GCT CAC TCT CAC CAC C-3′/IPSE-alpha 1-rev: 5′-ACA GTA TGT CCT TCT CCG TTT CGG T-3′, Kappa-5-for: 5′-TCG ATG GTT GAA CGG TTC GGA TG-3′/Kappa-5-rev: 5′-TGA CCT ACA GTC AAC CTC GGC T-3′, and Omega-1-for: 5′-GGA CGG AGA GGG ATG TAT CA-3′/Omega-1-rev: 5′-TTC CAA GGA ACG GGC AGT-3 that were published before [28,29,30].

### 2.7. Next-Generation Sequencing

High-quality RNA from GRX cells was isolated by an established procedure using CsCl gradient centrifugation. In brief, confluent cells were lysed and homogenized in a guanidine thiocyanate-containing buffer and the solution layered onto a cesium chloride cushion and centrifuged for 21 h at 21 °C and 25,000 rpm in a Beckman SW41 rotor. The RNA was diluted in sterile water, precipitated with ethanol, and finally resuspended in sterile water and the concentration and purity determined by UV spectroscopy. The quality of the purified RNA was further analyzed by standard gel electrophoresis in 1.2% (*w/v*) agarose/2.2 M formaldehyde gels in a buffer containing 20 mM 3-(*N*-morpholino)propanesulfonic acid (pH 7.0), 5 mM sodium acetate and 1 mM ethylenediaminetetraacetic acid (EDTA) and stained with ethidium bromide. The quality of RNA was further documented by using the Agilent 4200 TapeStation system (Agilent, Waldbronn, Germany). Subsequently, the mRNA was converted into a library of template molecules and prepared for subsequent cluster generation and DNA sequencing. Therefore, the NEBNext^®^ ultra^TM^ directional RNA library prep kit for Illumina (#E7765, New England Biolabs, Frankfurt am Main, Germany), the TruSeq RNA Single Indexes Set B kit (Illumina, San Diego, CA, USA), and the MiSeq platform MiSeq reagent kit v2 (300-cycles) (Illumina) by using manufacturer’s protocols. The final NGS data files were converted into fastq files. Construction and sequencing of the cDNA library was done in the IZKF genomic facility of the RWTH University Hospital Aachen.

### 2.8. Western Blot Analysis

Protein extracts were prepared from GRX and AML12 cells following standard protocols. Equal amounts of proteins (40 µg) or supernatants (21 µL) were heated at 80 °C for 10 min and separated in 4–12% Bis-Tris gels (Invitrogen, Darmstadt, Germany) under reducing conditions by using MES running buffer and electro-blotted on nitrocellulose membranes (Schleicher & Schuell, Dassel, Germany). Equal protein loading was monitored in Ponceau S stain and unspecific binding sites were blocked in TBST (10 mM Tris/HCl, 150 mM NaCl, 0.1% (*v/v*) Tween 20, pH 7.6) containing 5% (*w/v*) non-fat milk powder. The membranes were subsequently probed with antibodies specific for caveolin-1, collagen type I, collagen type III, collagen type IV, desmin, E-cadherin, fibronectin, HNF-α, IL-6, transferrin, vimentin, α-SMA, α-tubulin, β-actin, and GAPDH. The presence of MLV viruses in GRX cells was demonstrated by probing with a MLV (p30) specific antibody. To verify tetherin protein expression, membranes were incubated with specific anti-Bst-2 antibody. Primary antibodies were detected with horseradish-peroxidase (HRP)-conjugated secondary antibodies and the Supersignal™ chemiluminescent substrate (Perbio Science, Bonn, Germany). Information of all antibodies used in this study is listed in Appendix A.

### 2.9. RNA Extraction and PCR Analysis

The PureLink RNA Mini kit system, including DNase digestion, was used to isolate and purify total RNA from the different cell lines according to the manufacturer’s guidelines (all reagents from ThermoFisher Scientific). Superscript II reverse transcriptase and random hexamer primers (Thermo Fisher Scientific) were chosen to synthesize cDNA from 1 µg RNA as previously described [31]. Subsequently, the cDNA was used for reverse transcriptase (RT)-qPCR and RT-PCR. For RT-qPCR, cDNA (5 µL) was amplified in a total volume of 25 µL by using SYBR Green^TM^ qPCR SuperMix (ThermoFisher Scientific). For amplification of specific products, the following primers were used: mBst2-for: 5′- TCA GGA GTC CCT GGA GAA GA-3′, mBst2-rev: 5′-ATG GAG CTG CCA GAG TTC AC-3′, Gapdh-for: TGT TGA AGT CAC AGG AGA CAA CCT-3′, and Gapdh-rev: 5′-AAC CTG CCA AGT ATG ATG ACA TCA-3′, respectively. The thermal cycling conditions were set as follows: First, initial denaturation at 95 °C for 10 min, then amplification in 40 cycles of 95 °C for 15 s and 60 °C for 1 min. RT-qPCR was prepared in triplicates, performed in technical duplicates and expression of *Bst2* was normalized to *Gapdh* expression. Relative levels of *Bst2* were calculated by using the 2^−ΔΔCT^ method [32], and relative mRNA expression levels were represented as the normalized quantity of *Bst2* mRNA of each cell line in relation to the normalized quantity of *Bst2* of AML12 cells.

For conventional RT-PCR, cDNA was synthesized as described above and subjected to the following cycle conditions: 5 min initial denaturation at 95 °C, 1 min at 95 °C, 1 min annealing at 62 °C (*Bst2* #35 cycles) or 64 °C (*Gapdh* #20 cycles), 3 min extension at 72 °C, and final elongation at 72 °C for 10 min. The calculated sizes of amplicons generated by the primer combinations given above were 159-bp for *Bst2* and 432 bp for *Gapdh*. Amplified PCR products were separated in a 1.6% agarose gel by using TAE buffer and extracted for sequencing.

### 2.10. Rolling-Circle-Enhanced-Enzyme-Activity Detection (REEAD)

REEAD detection for active human immunodeficiency virus (HIV), murine leukemia virus (MLV), and mouse mammary tumor virus (MMTV) integrase were essentially done as described before by using the virus substrates listed in Appendix A [33]. As negative controls, we tested in parallel for the human immunodeficiency virus (HIV) integrase activity. The expression and purification of recombinant MLV and HIV integrases were essentially done as previously described [34]. For the REEAD analysis, GRX cell supernatants and culture medium alone were filtered through a 0.45 µM filter. To lyse the virus particles, 30 mM NaCl, 25 mM 4-(2-hydroxyethyl)-1-piperazineethanesulfonic acid (HEPES) pH 7.6, 0.1% Triton X-100, and 1 mM DTT was added, and the cell supernatants and medium were incubated 15 min on ice before being used for the REEAD analysis. Tests that contained the small-molecule drug raltegravir that inhibits the retroviral integrases of HIV and MLV were carried out in parallel [34].

## 3. Results

### 3.1. Phenotypic Characterization of GRX Cells

GRX cells are anchorage-dependent cells, which were established from fibrotic granulomas induced in C3H/HeN mice liver by infection with *S. mansoni* [5]. They have a highly proliferative phenotype. Compared to AML12 cells that originate from hepatocytes and have a flat, polygonal phenotype with round nuclei and granular cytoplasm [21], GRX cells acquire a more distinct fibroblast-like morphology when cultured on uncoated plastic (Figure 2A). In addition, GRX cells frequently contain lipid droplets that are visible when cultured at lower density. As cell lines that originate from connective tissue, GRX cells express extracellular matrix proteins such as collagen type I, collagen type III, collagen type IV, vimentin, and fibronectin (Figure 2B). In addition, GRX cells express α-smooth muscle actin (α-SMA), indicating that the cells are derived from HSCs and have acquired an activated phenotype. In addition, a Western blot analysis showed that GRX cells are also positive for Caveolin-1 (Cav-1), α-tubulin, and E-cadherin, while they showed no expression of hepatocyte nuclear factor 4 (HNF4-α). However, GRX cells showed strong expression of IL-6, a finding that was previously reported by others [35].

Moreover, the expression of typical HSC markers including fibronectin, Cav-1, collagen type IV, vimentin, collagen type I, and α-SMA that are also found in other HSC lines indicate that the cells most likely originate from the HSC lineage (Appendix A).

Electron microscopic analysis revealed the typical heterochromatic nuclei and the appearance of fat droplets within the cytoplasm (Figure 3A–C). Most strikingly, the cells contained spherical enveloped particles with an average diameter between 100 and 200 nm characteristic for retrovirus particles [5]. These particles were found in high concentration in the cytoplasm and attached to the cell surface (Figure 3D–F).

It appears that viral particles formed within these GRX cells are first enriched into groups, rerouted to the cytoplasmic membrane, released by budding forming typical protrusions, and stay attached at the surface of the cells (Figure 4).

### 3.2. Determination of Virus Identity

To determine the identity of the retroviral particles, we performed next-generation sequencing (NGS) of mRNA isolated from cultured GRX cells grown to confluence. Sequencing data received revealed that sequenced cDNA library contained large number of sequencing reads belonging to the *Mus musculus* endogenous virus ecotropic murine leukemia virus (MLV) [36]. Moreover, we were able assemble together the complete MLV sequence from overlapping NGS sequencing reads with a query cover of 100% and a sequence identity of 99.35% (not shown), suggesting that the retroviral particles produced by GRX are type VI retroviruses belonging to the gammaretroviral genus of the *Retroviridae* family. However, the NGS data revealed no indication for the presence of MMTV infection. To further confirm that the produced viruses are MLV, we next performed a Western blot analysis by using GRX protein cell extracts, and an antibody directed against MLV gag protein p30 (Appendix A).

Interestingly, the MLV (p30) protein was only detectable in the cell extracts of GRX cells and not in the supernatants of cultured GRX cells, suggesting that the release of the enveloped virus is blocked. However, it should be noted that the electron microscopic analysis showed large numbers of viral particles outside the cells and not attached to the cells. This is most likely due to the fact that the cells were rigorously scraped from the culture plate and intensively pipetted before being fixed for electron microscopic analysis.

Many previous reports have shown that the release of a broad spectrum of enveloped viruses is significantly blocked in the presence of tetherin also known as bone marrow stromal antigen 2 (Bst-2) [37,38,39]. As such, “physical tethering” by tetherin prevents effective secretion of viral particles and restricts cell–cell transmission of viruses. To test if tetherin is expressed in GRX cells, we extended our Western blot analysis and analyzed if GRX expresses tetherin (Figure 5). In this analysis, we used AML12 and the two murine breast cancer cell lines E0771 and 4T1 that are known to express large quantities of tetherin as positive controls [40,41].

This analysis revealed that GRX expresses tetherin mRNA (Figure 5A,B). The identity of the amplicons was further verified by sequencing (not shown). Next-generation sequencing of GRX mRNA further revealed that the respective mRNA carried only one silent mutation (TTC→TTT) at position 525 when compared to *Bst2* mRNA which is deposited in the GenBank nucleotide database (access. no. NM_198095.3). Moreover, in line with previous reports, tetherin showed the typical protein migration pattern in SDS gel electrophoresis (Figure 5C), which is due to extensive *N*-linked glycosylation known to be a prerequisite for tetherin antiviral activity [39].

### 3.3. Analysis Retroviral Activity

We next used REEAD technology to clarify if the supernatant from cultured of GRX cells contain active MLV integrase activity. The respective analysis showed no indication of MLV integrase in the supernatant of GRX cells (Figure 6). In line with the absence of MLV p30 protein (cf. Appendix A), this analysis revealed that active viral particles were absent in conditioned media of GRX cells, again underpinning the concept of ”physical tethering” of viral particles at the cellular surface.

### 3.4. Genotypic Characterization of GRX Cells

There is widespread agreement that studies performed with misidentified cell lines add misinformation to the literature, are likely to produce irreproducible biomedical research results, and can provoke additional studies of questionable value [42]. Although the GRX cell line has proven to be an effective experimental tool for many studies, a detailed genotypic characterization of this cell line or methods that allow authentication of this cell line has yet to be performed.

#### 3.4.1. Karyogram Analysis of GRX Cells

To characterize GRX cells genetically, we first performed chromosome G-banding on metaphase chromosomes and analyzed respective Giemsa stain by light microscopic analysis. This analysis showed that the karyotype of GRX cells is highly diverse. In representative karyograms, we observed changes in chromosome numbers and lengths (Figure 7). Increased numbers of chromosomes were frequently found for Chr 2, Chr 3, Chr 6, Chr 10, Chr 11, Chr 14, Chr 15, and Chr 19, whereas lower numbers of chromosomes were somewhat less frequently found for Chr 4, Chr 5, Chr 12, and Chr 16. Moreover, the karyogram of cells often showed a large number of chromosomal derivatives that could not be properly assigned to any chromosome.

In addition, many cells showed more than two complete sets of chromosomes, some with as many as 90 chromosomes (Appendix A).

#### 3.4.2. Chromosomal Arrangements in GRX Cells

In general, the house mouse usually shows a higher magnitude of Robertsonian (Rb) translocations than most other mammals and has therefore been proposed as a particularly amenable model to study this mutational process [43]. It was proposed that the very high frequency of Rb translocations in the ancestral karyotype of the house mouse that is composed of 40 acrocentric chromosomes, could be caused by “inherent genomic traits” such as the clustering of heterochromatic regions, the homology of pericentromeric satellite DNAs (satDNA), and the nicking activity of specialized protein binding to defined regions within satDNA sequences [44]. In particular, the satDNA sequences representing large tandem arrays of repeated motifs with high intraspecific sequence identity and often opposite orientations have been considered as key features promoting the formation of centric rearrangements [45].

Detailed analysis of GRX cells showed that Rb translocations (Appendix A) and many other chromosomal arrangements (Appendix A) occur at high frequency in GRX cells. Chr 19, Chr 18, and Chr 15 are frequently attached to themselves through their short arms. In addition, we observed interchromosomal fusions between Chr12, Chr 6, Chr 8, Chr 2, and Chr 3 with other chromosomes. Furthermore, we frequently found several derivatives, double minutes, quadriradial arrangements, and marker chromosomes (Appendix A).

#### 3.4.3. Short Tandem Repeat Analysis in GRX Cells

The karyotype analysis showed that GRX cells have a complex karyotype with several numerical and structural abnormalities, which occurred with variable pattern. To establish a simpler characteristic to uniquely identify GRX cells, short tandem repeat (STR) DNA profiling was performed by using the consensus 18 polymorphic markers proposed by the Consortium for Mouse Cell Line Authentication [24], which are also implemented in the Cellosaurus cell line knowledge resource [46].

The resulting PCR products were analyzed on a genetic analyzer resulting in electropherogram profiles with characteristic peaks for each highly polymorphic STR tested (Figure 8).

A search at the CLASTR database showed that the obtained STR profile was unique for GRX cells. Highest similarities regarding these 18 markers were found to EOC 20, HNOS, A-9, P19, and NCTC clone 929, respectively (Table 1).

### 3.5. Considerations on Biological Safety Classification of GRX Cultures

It is well accepted that cell cultures have become very beneficial as test systems for diverse applications in biotechnology and biomedical research. However, based on their potential harm to human health and the environment, a precise biological risk assessment of each cell line is required in most countries. Beside the characterization of intrinsic properties such as genetic modifications, harm may occur by potential contaminations within the culture. As mentioned above, GRX cells were established from granulomas induced in C3H/HeN mice liver by experimental infection with *Schistosoma mansoni* [5]. Moreover, in the first characterization of this cell line, transmission electron microscopic analysis revealed that this cell line produced viral particles of retrovirus type that was observed in the cytoplasm, inside small vesicles, or in spaces belonging apparently to a system of deep furrows in the cell cytoplasms [5].

Clearly, both contaminations with *S. mansoni* and retroviruses would be significant factors that need to be taken into consideration when estimating the biosafety risk assessment for these cells. In addition, such contaminations need particular attention because they may induce cytopathic effects that might limit the usability of the respective cell lines for several research questions.

*S. mansoni* is a human parasite provoking schistosomiasis. As such, exposure by inhalation, contact or absorption through needle stick or laceration might result in exposure incident and illnesses. Therefore, *S. mansoni*, *S. haematobium*, *S. intercalatum*, *S. japonicum*, and *S. mekongi* were classified in the Ordinance of the Federal Minister of Labor, Health and Social Affairs on the protection of employees against risks from biological agents that is valid in Germany into biosafety 2 [48]. Moreover, following these legally binding requirements, the handling of amphotropic (replication-competent and replication-defective) retroviruses including the infection of cells of risk group 1 is to be assigned to safety level 2. Similar regulations are also applicable in many other countries inside and outside of Europe. Nevertheless, the biosafety classification specified by the Banco de Células do Estado do Rio de Janeiro is level 1, suggesting that this cell line does not include contaminants causing human or animal disease [49].

To prove that GRX cells do not produce *S. mansoni* or retroviruses any longer, we next performed a Western blot analysis, immunocytochemical analysis, and expression analysis to exclude for potential contaminations that would be indicative for the presence of any life cycle forms of *S. mansoni.* Active infection with *S. mansoni* can be best demonstrated by screening for the presence of regurgitated worm antigens such as genus-specific polysaccharide structures [50]. In particular, the glycan-binding antibodies binding to a broad range of *S. mansoni* surface proteins containing cercarial O- and GSL-glycans have been found to be an extremely useful diagnostic for the detection of respective infections [51]. Immunocytochemistry using well-established monoclonal antibodies directed against schistosome gut-associated circulating anodic antigens (Figure 9A) or structures containing cercarial O- and GSL-glycans and/or cercarial *N*-glycans containing abundantly fucosylated glycans (Figure 9B) demonstrated that respective structures were absent in GRX cultures. In addition, we were able to demonstrate by RT-qPCR that specific mRNAs for the major antigenic glycoproteins IPSE/alpha-1, Kappa-5 and Omega-1 were absent in GRX (Figure 9C,D), again demonstrating that GRX cells lack glycan antigens that would indicate the presence of *S. mansoni*.

## 4. Discussion

Cell lines are extensively used as in vitro model systems in biomedical research. However, cell line authentication has received little attention in the past. There are many causes and scientific effects of misidentified and cross-contaminated cell lines giving rise to thousands of misleading and potentially erroneous papers [52]. There are efforts to establish guidelines for the use of cell lines in biomedical research [53]. New regulations in animal welfare and constant increase in the cost of animal experiments will guide many researchers to move one step back and work with cell lines. Moreover, many journals and agencies now recommend or require cell line authentication before they grant approval or acceptance of studies for publication.

In the field of hepatology, there are a number of spontaneous or experimentally derived immortalized HSC lines available that are widely used in many laboratories [4]. It is generally accepted that these cells are suitable experimental tools for analyzing different aspects of HSC biology. Immortalized cell lines grow continuously, have an almost unlimited lifespan, and based on their clonal origin should have a homogenous and specific phenotype. However, presently only the human HSC line LX-2 is genetically characterized and a STR profile for cell authentication available [54].

In the present study, we have established an STR profile for the murine HSC line GRX that relies on 18 highly polymorphic tetranucleotide repeats. The determined profile is unique and allows easy and fast authentication of GRX cells. Compared to other STR profiles deposited in the Cellosaurus database, the highest overlap was found with non-HSC lines EOC 20, HNOS, A9, P19, and NCTC clone 929. Like GRX cells, four of these murine cell lines were also established from mouse C3H strains but had a completely different origin and generation history. The first cell line (EOC 20) was originally established from a panel of 20 non-virus-transformed cell clones derived from individual microglial precursors from 10-day-old female C3H/HeJ [55]. The second line (i.e., A9) is an adherent connective tissue cell line established as a 8-azaguanine resistant cell line derived from mouse fibroblasts (L cells) [56], which originally were generated from an explant of subcutaneous connective tissue taken from 100-day-old male C3H/Andervont substrain mouse (i.e., the strain L cells) [57]. The third line (i.e., NCTC clone 929, also termed L-929) is a subclone of parental L strain originally derived from normal subcutaneous areolar and adipose tissue of 100-day-old male C3H/An mouse [58,59]. The fourth line (i.e., P19) is an adherent cell line with an epithelial morphology originally derived from a teratocarcinoma formed following extra-urine transplantation of egg cylinders isolated from female mice 7 day after copulation into the testis of 2- to 4-month-old C3H/HeHa mice [60].

HNOS is a highly metastatic cell line that was originally thought to originate from a human oral cavity squamous cell carcinoma [61]. However, based on isoenzyme analysis and a lack of amplification using human STR profiling, this cell line was later shown by the International Cell Line Authentication Committee to be misidentified and of murine origin [62].

GRX cells express typical protein markers of HSCs including α-SMA, Fibronectin, and different types of collagens suggesting that they have physical characteristics of fibroblasts. Consistently, they do not express the hepatocyte nuclear factor 4α (HNF-4α), which is a hepatocyte-specific gene involved in regulation of glycolysis, gluconeogenesis, fatty acid metabolism, and bile acid synthesis [63]. As such, GRX cells should have the proposed characteristics of mesenchymal connective tissue belonging to the smooth muscle cell lineage [5].

In our study, we found that GRX cells carry a high variety of numerical and structural chromosomal abnormalities. We frequently identified cells in our culture with increased numbers of chromosomes, Rb translocations, quadriradial arrangements, and fusions of different chromosomes. In rare cases, we further found pulverized chromosomes representing a visual phenomenon of apparent discontinuous DNA fragments on chromosome spreads caused by mitotic errors or DNA damage [64]. Such a genomic catastrophe also termed chromothripsis can be induced by activated retrotransposons or retroviruses that might provoke nonallelic homologous recombination when occurring in high copy numbers [65].

The electron microscopic analysis showed that GRX cells have the capacity to produce and secrete large quantities of retroviral particles. We initially thought that this retrovirus is MMTV because it is well-established that some inbred mouse strains, and in particular derivatives of the C3H strain, harbor either congenitally or genetically transmitted endogenous MMTV viral genomes [66].

Next-generation sequencing and Western blot analysis using an antibody directed against the MLV gag p30 protein revealed that GRX cells produce retroviral particles belonging to the gammaretroviral genus of the *Retroviridae* family. However, the REEAD analysis performed showed that no viruses with active integrase can be detected in the cell culture supernatant. Integrase enzymes are found in all retroviruses and are highly crucial for the process of viral integration into host DNA and virion maturation [67]. The finding that our MLV-specific integrase probes were unable to detect MLV-integrase in supernatants of cultured GRX cells might indicate that the produced viral particles strongly adhere to the surface of the cells. Such a block in the release of enveloped nascent virions at the cell surface was already reported for many viruses [37,38,39]. In line with the “physical tetherin” concept, we showed that GRX express tetherin both at the mRNA and protein level.

In addition, we showed that GRX cells lack any gut-associated circulating anodic antigens and cercarial O- and GSL-glycans derived from *S. mansoni* that was used to infect livers from which the GRX cells was established. This finding is of fundamental importance because *S. mansoni* is a human trematode parasite provoking schistosomiasis. *S. mansoni* is a biosafety level 2 organism for which special containment equipment and facilities are recommended for laboratory work [68]. The lack of specific glycostructures of *S. mansoni* in GRX cells that play important roles at the host–parasite interface [69,70] make these requirements superfluous and allow for work in a standard biosafety level 1 facility.

In sum, we have shown that GRX cells have the capacity to produce typical extracellular matrix components that are specific for HSCs. The cells have a highly variable karyotype with many numerical and structural aberrations. However, STR profiling is an effective method to authenticate GRX cells. In culture, GRX cells produce and secrete large quantities of retroviruses, but lack specific glycostructures that are relevant for parasite biology and host–parasite interactions.

Nowadays, several continuous growing HSC lines from mouse, rat, and humans are available [4]. However, only a few lines have found widespread use. Extensive literature is only available for the two rat lines HSC-T6 and CFSC (and its derivatives), the two human cell lines LX-2 and LI-90, and the murine cell line GRX investigated here. However, a STR profile for cell authentication was only reported before for LX-2 cells [54]. The definition of a STR profile for GRX that we have established in this study will now allow recognizing cross-contamination and misidentification when working with this cell line.

## 5. Conclusions

In this study, we have established a unique STR profile for murine HSC line GRX that allows for cell line authentication. Because this cell line is used in many basic studies investigating aspects of liver pathogenesis, this information is essential to identify cell misidentification and cross-contamination. The genetic information provided will make this cell line a more attractive cell line for biomedical research and helps to fulfill the request of fund agencies and scientific journals to incorporate authentication testing when working with immortalized cell lines to prevent misleading or false data, save money, and finally help to save scientific reputation. We further demonstrated that GRX cells produce MLVs but found no active viruses in the cell culture supernatant as indicated by a lacking of retroviral integrase activity. Moreover, *S. mansoni*-specific glycostructures are absent in GRX cells, which is relevant for classification of this cell line in biosafety level 1.

## Figures and Tables

**Figure 1 cells-11-01504-f001:**
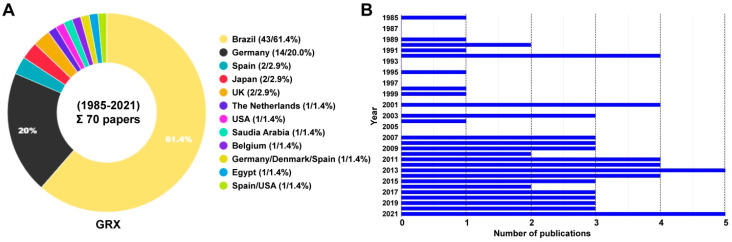
Usage of GRX cells during the period from 1985–2021. (**A**) A total of 70 studies using GRX cells were identified in a PubMed or Google search using search terms “GRX cells” or “GRX and hepatic”. Papers were excluded that used the term ‘GRX’ for other purposes (see Section 2 for details). In total, 69 papers were identified that used GRX cells. Most studies with these cells were conducted in Brazil, Germany, and Spain. The search was done on 5 November 2021. (**B**) Since the first report on GRX cells in 1985, reports using this cell line as an experimental tool steadily increased from year to year. For references on individual studies performed with GRX cells, refer to Appendix A.

**Figure 2 cells-11-01504-f002:**
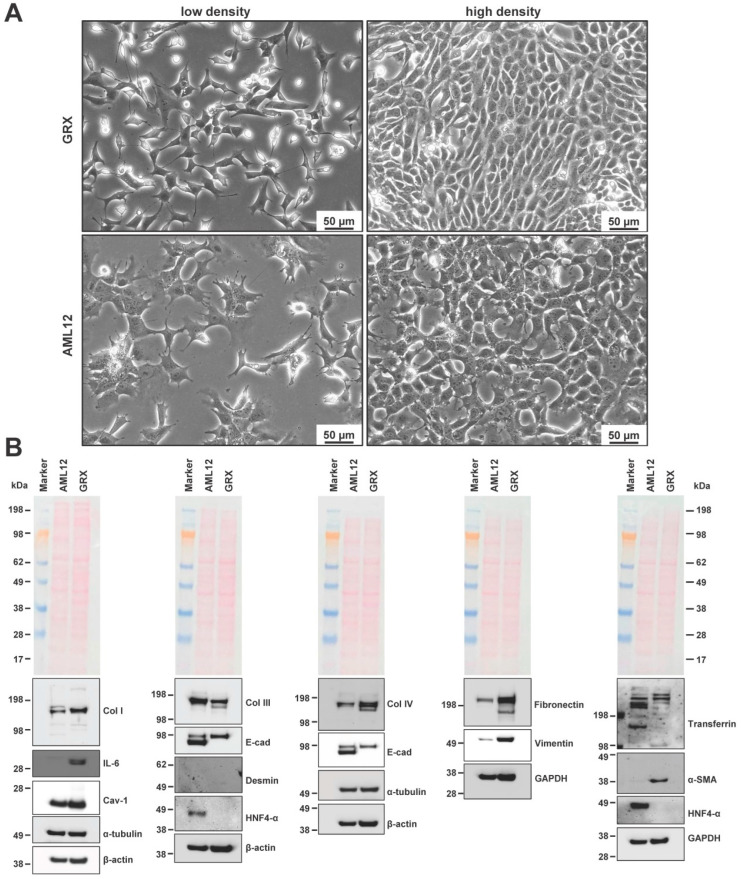
Phenotypic characteristics of GRX cells. (**A**) GRX and AML12 cells were seeded in cell culture dishes and representative images taken from subconfluent and confluent cultures. Original magnifications are 200×. The space bars correspond to 50 µm. (**B**) Cell extracts were prepared from AML12 and GRX cells and analyzed for expression of α-smooth muscle actin (α-SMA), caveolin-1 (Cav-1), collagen type I (Col I), collagen type III (Col III), collagen type IV (Col IV), desmin, E-cadherin (E-cad), and fibronectin, hepatocyte nuclear factor 4 (HNF4-α), interleukin 6 (IL-6), transferrin, and vimentin. The expression of α-tubulin, β-actin, or GAPDH and the Ponceau S stain were included to demonstrate equal protein loading.

**Figure 3 cells-11-01504-f003:**
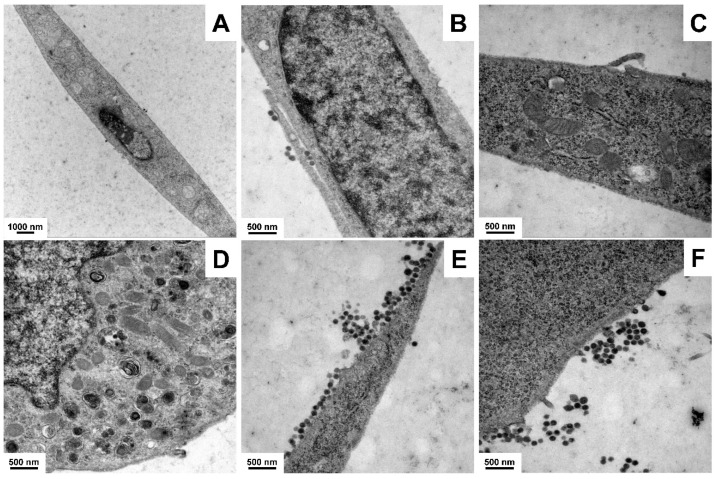
Representative electron microscopic analysis of GRX cells. Typical ultrastructural features of GRX cells are depicted including (**A**) overview of cell structure, (**B**) dense nucleus, (**C**) mitochondria and intracellular fat droplets, (**D**) intracellular retroviruses, and (**E**,**F**) retroviral particles at cell border. Magnifications: (**A**) 6000×, (**B**–**F**) 21,560×.

**Figure 4 cells-11-01504-f004:**
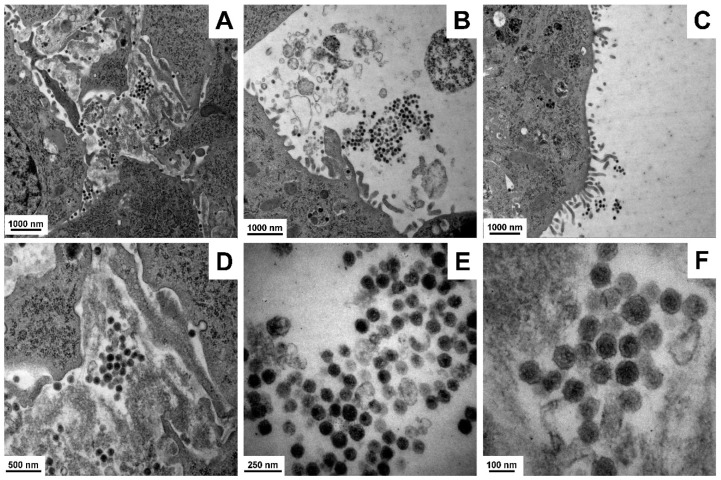
Retroviral load in GRX cells. (**A**–**F**) Representative images of retrovirus in GRX cultures are depicted. Magnifications: (**A**,**C**) 10,000×, (**B**) 12,930×, (**D**) 27,800×, (**E**) 60,000×, and (**F**) 100,000×.

**Figure 5 cells-11-01504-f005:**
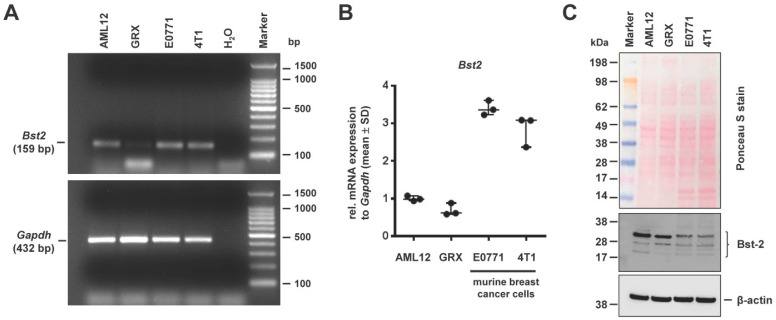
Expression of bone marrow stromal antigen 2 (Bst-2) in GRX. GRX cells were analyzed for Bst-2 expression by using AML-12 and murine breast cancer cell lines E0771 and 4T1 as positive controls. (**A**) Agarose gel electrophoresis of amplicons derived by RT-PCR showed a 159 bp product for *Bst2* in all analyzed cell lines. RT-PCR product of *Gapdh* (432 bp) served as control. (**B**) Quantitative mRNA expression analysis of *Bst2* by RT-qPCR verified expression in all cell lines. In this analysis, the relative mRNA expressions (measured in triplicates) were normalized to *Gapdh* expression. Values are given in relation to AML12 cells. (**C**) Western blot analysis confirmed Bst-2 protein expression. The Ponceau S stain is shown to demonstrate the integrity of proteins, and β-actin re-probing served as internal loading control. Please note, that for Bst-2 protein, more than one band (between 17–38 kDa) was visible, most likely due to post-translational modifications such as glycosylation.

**Figure 6 cells-11-01504-f006:**
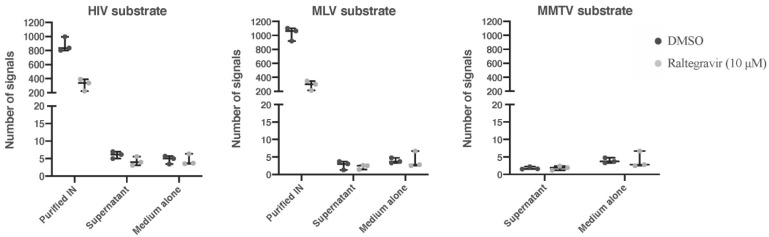
REEAD analysis for HIV, MLV, and MMTV integrases. The REEAD analysis for MMTV integrase was performed in supernatant of cultured GRX cells. As a positive control, purified integrase was analyzed in parallel. As negative controls, integrase activity was measured in buffer alone or in basal medium. As additional controls, tests that contained the integrase inhibitor raltegravir were carried out in parallel. Error bars represent standard deviations calculated from the results of three independent experiments.

**Figure 7 cells-11-01504-f007:**
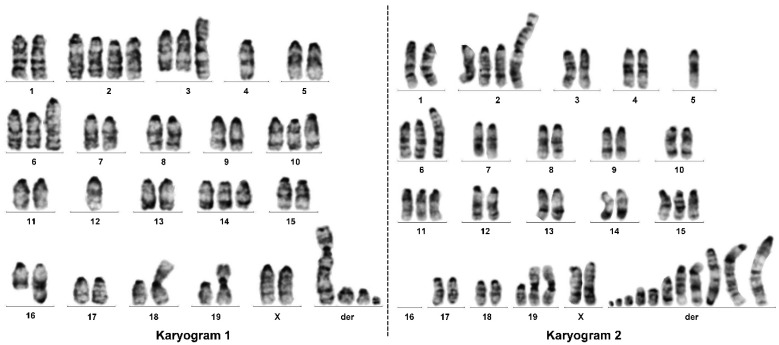
Representative karyograms found in GRX cells after Giemsa stain in light microscopic analysis.

**Figure 8 cells-11-01504-f008:**
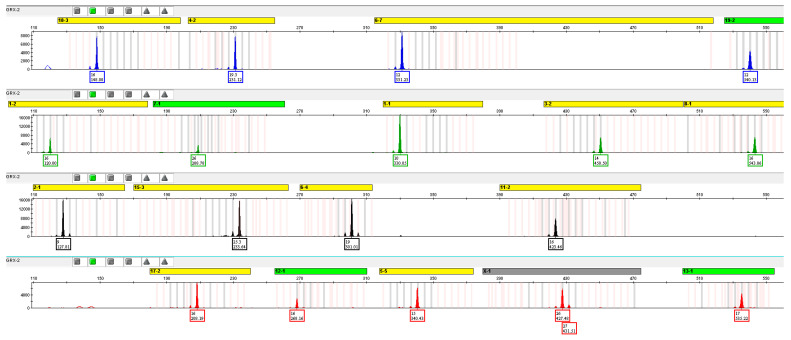
Extended electropherogram of GRX STR profile resulting from independent analysis with 18 markers. DNA from GRX cells were isolated and the genetic profile obtained by using the STR markers 1-1, 1-2, 2-1, 3-2, 4-2, 5-5, 6-4, 6-7, 7-1, 8-1, 11-2, 12-1, 13-1, 15-3, 17-2, 18-3, 19-2, and STR X-1, respectively.

**Figure 9 cells-11-01504-f009:**
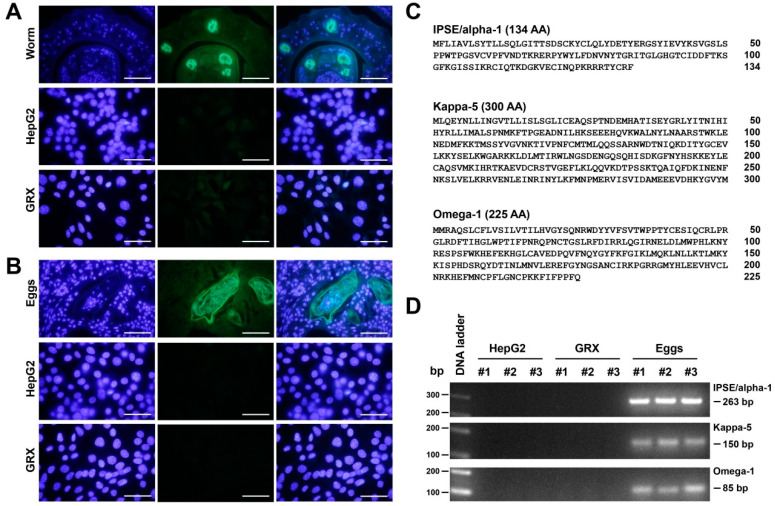
Analysis of potential *S. mansoni*-specific glycostructures in GRX cells. (**A**) The monoclonal antibody 54-4C2 produced against the schistosome gut-associated circulating anodic antigens (CAA) [50] was used to stain GRX cells (lower panel). HepG2 cells served as negative control (middle panel), while sections of the gut of the adult worm were used as a positive control (upper panel). (**B**) The glycan-binding monoclonal antibody 128-1E7 that binds with high affinity to a broad range of proteins containing cercarial O- and GSL-glycans and to a selection of cercarial *N*-glycans that all contain abundantly fucosylated glycans [51] was used to stain GRX cells. Although this antibody failed to recognize any epitopes on GRX (lower panel) and HepG2 (middle panel), the antibody-stained antigens that are expressed on the eggs in hepatic granuloma of bisex-infected mice (upper panel). Original magnifications in (**A**,**B**) are 1000×. The scale bars correspond to 50 µm. (**C**) Protein sequences of the major antigenic glycoproteins IPSE/alpha-1, Kappa-5 and Omega-1 from *S. mansoni*. Amino acid positions are given on the right margin. The protein sequences were taken from GenBank (https://www.ncbi.nlm.nih.gov/genbank/) entries deposited under access. nos. XM_018799269.1, AY903306.1, and DQ013207 (accession date: 15 April 2022). (**D**) RNA from HepG2 cells, GRX cells, and from *S. mansoni*-infected mouse liver was subjected to RT-qPCR. Specific amplicons for IPSE/alpha-1, Kappa-5 and Omega-1 were only produced from *S. mansoni* infected mouse liver. Primer combinations used for this analysis are given in the Material and Method section (see Section 2.6).

**Table 1 cells-11-01504-t001:** Matching of the 18 short tandem repeats (STRs) obtained for GRX cells to known STR profiles listed in Cellosaurus database *.

STR Marker	GRX	EOC 20	HNOS	A-9	P19	NCTC Clone 929
1-1	10	10	10	10	10	10
1-2	16	16	16	17, 18	16, 17	17
2-1	9	9	9, 16	9	9	9
3-2	14	14	14	14	14	13, 14
4-2	19.3	19.3	20.3	20.3, 21.3	13, 21.3	20.3
5-5	15	15	15	14, 15	14	14
6-4	19	19	18	17, 18	18	17, 18
6-7	12	12	12, 16	12	12	12
7-1	26	26	26.2	26, 27	26	25, 26, 27
8-1	16	16, 17	16	15, 16, 17	16	16
11-2	16	17	16	16	16	15, 16
12-1	16	16	16,17	16	16	16
13-1	17	17	17	17	17	17
15-3	25.3	25.3	22.3, 25.3	25.3	26.3	24.3, 25.3, 26.3
17-2	16	16	15, 16	15	15	15
18-3	16	16	16	16	16, 17	16
19-2	12	12	12, 13	12	12, 14	12
X-1	26, 27	27	27	26, 27	26	26, 27

* The mouse STR markers of GRX from this second analysis were used to search the Cellosaurus database (release 38.0) by using the STR similarity search tool (CLASTR 1.4.4) at Expasy [25,47]. The STR profiles of cell lines EOC 20 (CVCL_5745), HNOS (CVCL_M838), A-9 (CVCL_3984), P19 (CVCL_2153) and NCTC clone 929 (CVCL_0462) showed the highest similarity, which were 89.47% (EOC 20), 69.77% (HNOS), 66.67% (A-9), 63.41% (P19) and 62.22% (NCTC clone 929), respectively.

## Data Availability

The STR data of GRX cells reported in this study were deposited in the Cellosaurus data base under access. no. CVCL_M115 [71]. In addition, complete NGS dataset in fastq format for GRX cells are available on request from the authors.

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
