# Peer review of "Genetic and Molecular Characterization of the Immortalized Murine Hepatic Stellate Cell Line GRX"

_cells, 2022, doi:10.3390/cells11091504_

Round 1

Reviewer 1 Report

GRX cells have been increasingly used or discussed as a suitable in vitro model for studying the cellular and molecular characteristics of hepatic stellate cell biology. In this study, the authors using Short Tandem Repeat (STR) Profiling, Next Generation Sequencing, Rolling-Circle-Enhanced-Enzyme-Activity Detection (REEAD), Karyotyping et al. establish a unique STR profile for GRX cells and further demonstrated that GRX cells produce MLVs, but found no active viruses in the cell culture supernatant as indicated by a lacking retroviral integrase activity. Moreover, the authors also validate that S. mansoni-specific glycostructures are absent in GRX cells, which is relevant for classification of this cell line in biosafety level 1. Their findings will help to fulfill the recommendations for cellular authentications required by many granting agencies and scientific journals when using GRX cells.  The definition of characteristic STR spectrum will increase the value of GRX cells in related research. 

Overall, this is an interesting study. But there are still several major concerns need to be clarified by the authors.

  1. As the authors states in the manuscript “the usage of cell lines is in general prone to misidentification and repeated passaging might provoke genotypic, karyotypic, or phenotypic drifts, which may result in cellular sublines that are phenotypically or genetically heterogenous”. Cell line passage number is an important consideration when designing an experiment. At higher passages, it is generally understood that cell health begins to decline and, when this occurs, the result can be variable data. The authors need to clarify how did the authors prepare GRX cells? How many passages the GRX cells been used in this study? Is there any difference between the low passages and higher number passages in the GRX cells?
  2. The authors need to explain that how they prepared the GRX cells supernatant and culture medium for the western blot and Rolling-Circle-Enhanced-Enzyme-Activity Detection (REEAD) et al. In Figures 3 and 4, the images clearly show that the virus particles are outside the cell and not attached to the cell, which contradicts the implication of Figure 5.
  3. In figure 7, what does the “purify IN” mean? The author had better clarify what each letter in GRX stands for.
  4. And in figure 11 A-B, the magnification is 1,000 x, But the image magnification (nuclear staining) doesn't seem to match. The same problem appears in the GRX high-density image in Figure 2A.
  5. It is better for the authors to ask English speaker or professional to revise the manuscript and make the grammar, sentence trends easier for reader to go through and understand.

Author Response

Dear Reviewer 1,

many thanks for reading our paper and your important comments. Our response to your comments are given in the attached pdf-file.

Regards

Ralf Weiskirchen

Reviewer 2 Report

The authors provided details of the murine cell line GRX, which would be helpful to the cellular authentications. However, GRX cells were not widely used. There are some other concerns:

  1. In the introduction, the authors are recommended to provide the background of hepatic stellate cells. The quiescent and activated state of HSCs.
  2. How about the other widely used rat HSCs cell line HSC-T6?
  3. There are too many figures in the manuscript.
  4. Have the author compared the GRX with other HSCs, like LX-2 cells in the phenotype?
  5. The manuscript do not belong to the Cellular Immunology section.

Author Response

Dear Reviewer 2,

many thanks for reading our paper and your important comments. Our response to your comments are given in the attached pdf-file.

Regards

Ralf Weiskirchen

Reviewer 3 Report

This is a very well written article on the characterization of the immortalized murine hepatic stellate cell line GRX, with the establishment of the STR profile. The authors showed a great overview of the use of the GRX cell line, showing 70 references. Overall, the paper is well written and up-to-date. However, it requires some additional westerns blots and representative images in the phenotypic characterisation of GRX cells, where it is missing a positive control of HSC already characterized, such as LX2 cell line, to compare with the GRX cell line. 

Author Response

Dear Reviewer 3,

many thanks for reading our paper and your important comments. Our response to your comments are given in the attached pdf-file.

Regards

Ralf Weiskirchen

Round 2

Reviewer 1 Report

Compared with the previous edition, this manuscript has obvious improvement in writing, method description and so on.  The author answered most of the reviewer's concerns and questions. There are a few minor points that need to be clarified.

  1. In Figure 5 and 6, All the bar chat figures best to replace with box-and-whiskers plots and the individual data points need added in plots.
  2. Need RT-qPCR IPSE/alpha-1, Kappa-5 and Omega-1 primer sequence information.

Author Response

Dear reviewer 1,

again many thanks for your input. Please see our response to your comments in the attached pdf-file.

Regards

Ralf Weiskirchen

(on behalf of all authors)

Reviewer 3 Report

The major concern has been corrected by adding a new Supplementary Figure 1.

Author Response

Dear reviewer 3,

many thanks for re-evaluating our paper. We greatly appreciate the time you spent.

Regards

Ralf Weiskirchen

(on behalf of all authors)